# Glutamatergic Signaling a Therapeutic Vulnerability in Melanoma

**DOI:** 10.3390/cancers13153874

**Published:** 2021-07-31

**Authors:** Kevinn Eddy, Suzie Chen

**Affiliations:** 1Graduate Program in Cellular and Molecular Pharmacology, School of Graduate Studies, Rutgers University, Piscataway, NJ 08854, USA; ke112@gsbs.rutgers.edu; 2Susan Lehman Cullman Laboratory for Cancer Research, Rutgers University, Piscataway, NJ 08854, USA; 3Rutgers Cancer Institute of New Jersey, New Brunswick, NJ 08901, USA; 4Environmental & Occupational Health Sciences Institute, Rutgers University, Piscataway, NJ 08854, USA

**Keywords:** melanoma, glutamatergic signaling, therapeutic targeting, Metabotropic Glutamate Receptor, glutamate, cancer mouse models, MAPK, PI3K/AKT, anti-glutamatergic signaling inhibitor

## Abstract

**Simple Summary:**

Out of all the skin cancers, melanoma is the most aggressive and dangerous form due to its high metastatic propensity. Patients with late-stage melanomas have poor prognosis as their five-year survival rate is only 27% while the survival rate for primary melanomas is 99%. Metastatic melanomas are resistant to most therapeutic approaches, progress quickly, and account for the majority of mortalities in melanoma patients. Melanomas like other cancers are driven by the dysregulation of the normal cellular networks that leads to uncontrolled cell proliferation, altered cellular metabolism and dissemination of tumor cells to distal organs. Our lab has described the oncogenic role of a normal neuronal receptor, Metabotropic Glutamate Receptor 1, that when aberrantly expressed in melanocytes leads to deregulated glutamatergic signaling in melanocytes. Activation of this receptor results in a cascade of disorders that promote cell transformation and tumor formation. Here we will explore the contribution of abnormal glutamatergic signaling to melanoma.

**Abstract:**

Like other cancers, melanomas are associated with the hyperactivation of two major cell signaling cascades, the MAPK and PI3K/AKT pathways. Both pathways are activated by numerous genes implicated in the development and progression of melanomas such as mutated *BRAF*, *RAS*, and *NF1*. Our lab was the first to identify yet another driver of melanoma, Metabotropic Glutamate Receptor 1 (protein: mGluR1, mouse gene: *Grm1*, human gene: *GRM1*), upstream of the MAPK and PI3K/AKT pathways. Binding of glutamate, the natural ligand of mGluR1, activates MAPK and PI3K/AKT pathways and sets in motion the deregulated cellular responses in cell growth, cell survival, and cell metastasis. In this review, we will assess the proposed modes of action that mediate the oncogenic properties of mGluR1 in melanoma and possible application of anti-glutamatergic signaling modulator(s) as therapeutic strategy for the treatment of melanomas.

## 1. Melanoma Statistics, Etiology and Biology 

Amongst all cancers, skin cancer is the most common and can be divided into basal cell carcinoma (BCC), squamous cell carcinoma (SCC), Merkel cell carcinoma (MCC), Kaposi sarcoma (KS), lymphoma of the skin and melanoma. Melanoma accounts for only 1% of all skin cancer cases but disproportionality accounts for the majority of skin cancer related deaths. This high mortality rate is attributed to its high metastatic propensity to migrate and colonize distal organs that include lymph nodes, lung, liver, bone, and brain [1]. In its early stages, melanoma in situ is “curable” with the five-year survival rate at 99%, however, once it is metastasized the survival rate drops to 27% [2]. In 2021, 106,110 new cases of invasive melanoma will be diagnosed in the United States with about 7180 patients expected to die [2]. These grave statistics reveal the importance of dissecting and understanding the complex circuitry of metastatic melanoma in order to develop novel therapeutic treatments for this deadly disease.

Melanoma is associated with various risk factors that include gender, fair skin, family history of skin cancer, age, UV exposure, and number of moles [1,3,4]. Men were shown to have worser outcomes in prognosis and survival when compared to women potentially due to sex hormones differences [1,5]. Inherited genetic defect(s) commonly associated with melanocytic transformation are the cell cycle regulating genes, *CDKN2A*, and *CDK4*, a gene associated with skin pigmentation, *MC1R*, and the genetic disorder xeroderma pigmentosum (XP), occurring as a result of mutations within the nucleotide excision repair (NER) machinery [6,7,8,9,10,11,12,13,14]. 

Melanocytes are found in various locations including the skin, mucosal tissue, uvea, heart, inner ear, and hair follicles. Based on the location of transformed melanocytes, melanoma can be categorized into cutaneous or non-cutaneous melanoma. Cutaneous melanoma is commonly associated with sun exposed locations and makes up the majority of cases, while non-cutaneous melanoma is found in sun-shielded areas [1]. Cutaneous melanoma can be divided into chronically sun-induced melanoma (CSID), non-chronically sun-induced melanoma (non-CSID) and the four genomic subtypes, *BRAF*, *RAS*, *NF1*, and triple wild-type [1,15]. CSID is commonly associated with individuals who are older than 55 years old, have a high mutational burden and contain genetic abnormalities in *neurofibromin 1* (*NF1*), *KIT*, *NRAS*, or *BRAF* non-V600E [16,17]. Non-CSID melanoma is found in areas that are intermittently sun-exposed, in patients who are younger than 55 years old, melanomas with moderate mutational burden and are dominated by *BRAF* V600E mutations suggesting that they originate from nevi [1,16,18]. The genomic subtypes of cutaneous melanoma will be discussed in the next section, “Genetics and Altered Signaling Pathways in Cutaneous Melanoma”. 

Non-cutaneous melanoma is a rare form of melanoma associated with a low tumor mutational burden, chromosomal gains/losses, and accounts for less than 10% of melanoma cases [1,19]. This subtype can be further subdivided into mucosal melanoma (1.3%), acral melanoma (2–3%) and uveal melanoma (5.2%) [1]. Uveal melanoma is frequently associated with mutations in the *CYSLTR2*, *PLCB4* and *GNAQ/GNA11* genes [20,21,22,23]. Like uveal melanoma, mucosal melanoma also harbors mutations in the *GNAQ/GNA11* genes, albeit rare, as well as others that include *KIT*, *CDK4*, *CCND1*, *CDKN2A*, *NF1*, *BRAF*, *NRAS*, *SF3B1*, *PTEN*, *TPR*, *SPRED1* and *TTN* genes [19,24,25,26,27,28,29,30,31]. Melanoma in non-hair bearing regions of the skin such as soles, palms and under fingernails/toenails are defined as acral melanoma and have the worst prognosis compared to other melanoma subtypes due to delay in diagnosis [32,33,34,35]. Genetic alterations frequently attributed to this subtype are: *PDGFRA*, *KIT*, *NF1*, *GNAQ1*, *NRAS*, *BRAF* and *TERT* promoter mutations [25,36,37,38,39]. The complex genetic makeup of the various melanoma subtypes converges onto two major signaling cascades, the MAPK and PI3K/AKT pathways [1]. Activation of the MAPK pathway leads to uncontrolled cell proliferation and stimulation of the PI3K/AKT pathway that potentiates tumor cell survival, growth signals and metastasis [40]. Other contributors that promote the uncontrolled proliferative capacity of melanoma cells include mutations in the cell cycle regulating genes, *CDKN2A* and *CDK4*, as well as promoter region of *Telomerase Reverse Transcriptase (TERT)* [3,16,19,24,26,41,42,43]. TERT, is a critical catalytic subunit in telomerase, an enzyme essential for the maintenance of telomeres and cellular senescence [42].

## 2. Genetics and Altered Signaling Pathways in Cutaneous Melanoma 

Acquisition of key altered genetic events are required for the successful transition of a normal epidermal melanocyte into melanoma in situ and eventually into metastatic melanoma. These key modified events are a result of inherited and/or acquired somatic mutations within key driver genes. Mutually exclusive mutations within the *BRAF* or *NRAS* (or *HRAS* albeit rare) genes in melanocytes leads to melanocyte hyperplasia and establishment of detectable and/or undetectable melanocyte nevi (benign tumor) [16,18,43,44,45,46,47,48,49,50,51]. To note, a significant proportion of melanomas do not arise from nevi [16]. This phase is termed as the initiation phase, where a melanocyte gains the capacity to become cancerous, however, it may be eliminated or undergo senescence if additional mutations are not acquired to transform it into melanoma in situ [16,43,49,52]. The breakthrough and invasive phase are characterized by the loss of *INK4a*, *ARF* and/or *PTEN* [44,49,53,54,55,56,57,58,59,60]. In addition to the loss of these tumor suppressor genes, the acquisition of the *TERT* promoter mutations is necessary for melanomas to become replicative immortal [16,24,41,42,43]. To note, a *KRAS* G12V mutant inducible mouse model was shown to develop melanocyte nevi that subsequently progressed into melanoma and had the intact tumor suppressor genes, *INK4a* and *TP53* [61]. However, the authors stated that additional mutations are required to induce melanocytic transformation into melanoma as the latency of melanocytic transformation has a median period of 4 months [61]. This phase is characterized by melanocyte nevi acquiring additional mutations to sustain the uncontrolled proliferative capacity thereby enabling them to transform into primary melanoma and eventually metastasize to distal organs [16,43,49,51]. Additional genetic aberrations that support the development of invasive melanoma are in the chromatin remodeling complex SWI/SWF components, and *TP53* genes [16,43,49,50]. To note, important considerations should be made in understanding the cooperative relationship between passenger and driver mutations since a cancer cell requires dysregulation of various genes/pathways to successfully become malignant [62]. By understanding this relationship, we can identify passenger mutations which may confer a melanoma (or other cancers) insensitive or resistant to targeted driver therapies and cells that harbor these mutations may become the dominant clone after treatment [62]. 

Clinically detectable primary and metastatic cutaneous melanoma can be sub-divided into four genomic classifications: *BRAF*, *RAS*, *NF1*, and triple wild-type [15]. The *BRAF* subtype is associated with BRAF hot spot mutations V600R, V600K, V600E, and K601E that are mutually exclusive with *NRAS* hot-spot mutations, while *BRAF* non-hotspot mutations co-occurred with *NF1* and *N/H/K-RAS* hot spot mutations [15]. This is the largest subtype identified in the Cancer Genome Atlas study, with 52% of the clinical samples testing positive for *BRAF* hot spot mutations [15]. The *RAS* subtype is comprised of hotspot mutations in the *N/H/K-RAS* genes [15]. The *NF1* subtype has the highest tumor mutational burden of all the four subtypes and half of those mutations have been associated with a loss of function mutation [15]. NF1 is a GTPase-activating protein, which regulates the MAPK pathway and has been shown to downregulate RAS activity [15]. In this subtype, mutated *NF1* is negatively correlated with hot-spot mutations in the *BRAF* gene but not with hot spot *RAS* mutations [15]. The last subtype identified in this study is the triple wild-type that does not harbor mutated hot spot mutations in the *NF1*, *BRAF* or *N/H/K- RAS* genes [15]. In the triple wild-type subtype there are rare low frequency driver mutations identified, *KIT*, *GNAQ*, *GNA11*, *EZH2*, and *CTNNB1* [15]. In the Cancer Genome Atlas study, no comment was made on the relationship between the genomic subtypes identified in their study and its relation to CSID and non-CSID melanoma subtypes. Albeit not tested directly, the triple wild-type genomic subtype seems to fall into the non-CSID melanoma as only 30% of the samples harbored UV signatures while the other three subtypes seem to fall into the CSID melanoma subtype with the *BRAF* subtype harboring 90.7% UV signatures, *RAS* subtype with 93.5% UV signatures and *NF1* with 92.9% UV signatures [15]. Based on these genomic classifications it has been suggested that patients with *BRAF*, *RAS* and *NF1* subtypes may derive benefit from MAPK inhibitors while triple wild-type will benefit from receptor tyrosine kinase inhibitors (RTKS) [15]. Interestingly all four genomic subtypes of cutaneous melanoma are associated with aberrant activation of the MAPK and/or PI3K/AKT pathway which supports tumor cell growth, proliferation, survival, and anti-apoptosis signals [15,63]. As “OMICS” studies become more feasible and routine in the clinics, these four genomic melanoma subtypes will likely have to be redefined and reclassified as the complexities of melanoma biology will be further unraveled. In line with this, our lab has uncovered the complex role of a normal neuronal receptor, Metabotropic Glutamate Receptor 1 (protein: mGluR1, mouse gene: *Grm1*, human gene: *GRM1*) in melanoma development and progression [64]. The oncogenic activities of mGluR1 expressing melanoma cells are independent of *BRAF* and *NRAS* mutations; additionally it has been found that a polymorphism in the *GRM1* gene is predominately found in non-CSID melanoma patients [65,66,67]. Clinical data acquired from cBioPortal for Cancer Genomics stated that in human melanomas the frequent alterations found in the *GRM1* gene are results of mutations, amplifications, and/or deletions, with desmoplastic melanoma being dominated with mutations in the *GRM1* gene (Figure 1A) [68,69]. Furthermore, it has been shown that *GRM1* co-occurs with *NF1* or *BRAF* alterations while *NRAS* is mutually exclusive (Figure 1B) [68,69]. However, we have shown co-occurrence of mGluR1 expression and mutated NRAS in human melanoma cell lines and patient biopsies [66,68,69,70,71].

## 3. Glutamatergic Signaling in Melanoma Pathogenesis 

### 3.1. Discovery of Metabotropic Glutamate Receptor 1 as a Driver in Cancer 

Our lab was the first to show the oncogenic properties of ectopic mGluR1 expression in melanoma development and progression. We uncovered the previously unknown property of mGluR1 in vivo from a classic case of insertional mutagenesis mediated by a 2 kilobase (kb) genomic DNA fragment, clone B, that was shown to commit fibroblasts to undergo adipocyte differentiation in vitro (Figure 2A) [72,73]. One of the five transgenic founder mice derived from clone B, TG-3, showed pigmented lesions that was subsequently confirmed to be melanoma (Figure 2A) [74,75,76]. We went on to show that clone B inserted into the intron 3 sequence of the *Grm1* gene lead to the concurrent deletion of a 70 kb DNA sequence resulting in spontaneous melanoma development in TG-3 and its progenies (Figure 2A) [64,74,76]. To confirm that the aberrant expression of mGluR1 in melanocytes was the driver of this phenotype, we created another mouse model TG(*Grm1*)Epv, where *Grm1* cDNA was under the control of a melanocyte specific promoter, dopachrome tautomerase (Dct) (Figure 2B) [64]. TG(*Grm1*)Epv develops melanoma with similar onset and progression as the original TG-3, confirming the involvement of *Grm1* in melanomagenesis (Figure 2) [64]. Normal melanocytes do not express mGluR1, the ectopic expression of a normal neuronal receptor, mGluR1 in melanocytes leads to cellular transformation in vitro and spontaneous metastatic melanoma formation in vivo with 100% penetrance (Figure 2 and Figure 3) [64,66,67,74,76,77,78,79,80,81,82]. Furthermore, sustained mGluR1 expression is required for the maintenance and progression of this transformed phenotype in vitro and tumorigenesis in vivo [67,77,83]. We and others have shown that our *Grm1*-driven spontaneous melanoma prone mice develop both cutaneous and non-cutaneous melanoma (Figure 2 and Figure 3) [64,74,76,78,79,80,81,84]. Additionally, we have derived a spontaneous *Grm1* expressing amelanotic melanoma prone mouse model, LLA and a hairless spontaneous *Grm1* expressing melanoma prone mouse model, TGS (Figure 3) [78,79,80]. 

### 3.2. Oncogenic Signaling Mediated by Metabotropic Glutamate Receptor 1 

mGluR1 is a seven transmembrane G-protein coupled receptor (GPCR) that is activated by L-glutamate [40]. mGluR1 is normally expressed in the central nervous system and is involved in memory and learning [40,85,86]. Our findings that mGluR1 plays a role in melanomagenesis in mice prompted us to examine human melanoma cell lines and biopsies for mGluR1 expression. We first examined human melanoma cell lines and found that 23 of 25 lines were positive for mGluR1 expression [40]. We then examined 175 melanoma biopsies from primary to metastatic lesions and found approximately 60% of these samples expressed mGluR1 at the mRNA and protein levels [40]. Furthermore, Funasaka and colleagues have shown that 33.3% of nevi (common nevi, blue nevi, and spitz nevi) are positive for mGluR1 expression and 77.7% of metastatic melanomas are positive for mGluR1 expression [87]. In human and mouse melanoma cell lines, mGluR1 activation leads to similar downstream signal transduction as neuronal cells in the central nervous system (CNS) [88]. We speculate that since melanocytes are of neural origin, the basic machinery required for mGluR1 signaling is already present within melanocytes hence leading to oncogenic activities and transformation into malignant melanoma (Figure 4). We also demonstrated that in mGluR1 positive melanoma cells there is elevated levels of glutamate in the tumor microenvironment that contributes to the hyperactivation of mGluR1 (Figure 4) [66]. Activation of mGluR1 causes a conformational change within the extracellular domain of the receptor resulting in the intracellular G-protein subunit, G_α_, to exchange GDP for GTP and dissociation of G_αq/_G_α11_ with the G_βγ_ subunits (Figure 4) [40]. Dissociation of G_αq/_G_α11_ from G_βγ_ allows for the activation of phospholipase C (PLC) thereby cleaving phosphatidylinositol-4,5-diphosphate (PIP_2_) into inositol 1,4,5-triphosphate (IP_3_) and diacyl glycerol (DAG) (Figure 4) [40,89,90]. IP3 then diffuses from the inner cellular membrane into the cytosol while DAG remains attached to the membrane (Figure 4) [40,91]. IP3 interacts with the endoplasmic reticulum (ER) releasing calcium (Ca^2+^) into the cytoplasm (Figure 4) [40,90,92,93]. Increased Ca^2+^ levels in the cytoplasm and interaction of DAG with protein kinase C (PKC) activates PKC’s kinase activity there by activating RAS followed by the activation of the MAPK and PI3K/AKT pathways (Figure 4) [40,83,89,90,91,92,93,94,95]. Evidence from our lab suggests that mGluR1 mediated activation of the PI3K/AKT pathway might be a result of the transactivation of the insulin-like growth factor 1 receptor (IGF-1R) via Src (Figure 4) [96,97]. Furthermore, we also showed the importance of temporal expression of mGluR1 and mutated *BRAF (V600E)* in tumorigenesis [97]. Using the well-known *Tyr::CreER; BRaf^CA/+^* mice and topical administration of 4-hydroxytamoxifen led to the expression of mutated *Braf*, the appearance of highly pigmented lesions, and mGluR1 expression [97,98]. This was not sufficient to induce tumorigenesis even after 17 months and these pigmented nevi remained in senescence [97,98]. However, if mGluR1 expression occurs first in melanocytes, such as in TG-3 and TG(*Grm1*)Epv, it is sufficient to induce tumorigenesis regardless of *BRAF* genotype [97]. Taken together these results suggests that the ectopic expression of a normal neuronal receptor, mGluR1, in a neural crest originating cell, melanocytes, can induce melanocytic transformation into melanoma, mediated by the hyperactivation of the MAPK and PI3K/AKT pathways (Figure 4). 

In *GRM1/Grm1* driven melanoma mouse models we demonstrated correlations between larger tumors with prominent vasculature [77,99]. This is a result of mGluR1 activation of the PI3K/AKT/mTOR pathway upregulating HIF-1α expression resulting in enhanced secretion of angiogenic factors, VEGF, and interleukin-8 (IL-8) (Figure 4) [99]. Improved angiogenesis supports the growth of these rapidly proliferating cells by providing nutrients, and routes for tumor dissemination to distal organs. The Bosserhoff group used our *Grm1*-driven mouse model, TG(*Grm1*)Epv, to elucidate the function of the tumor suppressor gene, deubiquitinase cylindromatosis (CYLD), in melanoma development and progression [100]. CYLD is a deubiquitinating enzyme, which targets BCL-3, and loss of CYLD leads BCL-3 translocation into the nucleus and activation of genes involved in proliferation and metastasis [101]. *CYLD* −/− TG(*Grm1*)Epv mice showed early melanoma onset and tumor progression compared to *CYLD* +/+ TG(*Grm1*)Epv mice [100]. Loss of *CYLD* was shown to enhance the metastatic propensities of *Grm1* driven melanoma cells by increasing their migratory, angiogenic and vascular mimicry potential (Figure 4) [100]. This intriguing finding raises the question, are there differences in CYLD expression between mGluR1 positive and negative cells? Additionally, how does mGluR1 positive melanoma cells regulate *CYLD*, and what prompts the temporal inactivation of *CYLD* by mGluR1 expressing melanoma cells? Furthermore, we have shown that mGluR1 expressing melanoma cells in order to overcome apoptotic signals constitutively activate the transcription factor, NF-қB, that regulates cell survival and growth genes (Figure 4) [102]. 

Another mechanism attributed to the aggressive metastatic nature of mGluR1 positive melanomas is exosomes (Figure 4) [103]. Exosomes are small nano-sized vesicles (30–120 nm) that are comprised of lipids, nucleic acids, and proteins [98]. All cell types release them; however cancer cells release more exosomes than their normal counterparts [98]. Cancer exosomes are involved in promoting the formation of the pre-metastatic niche at distal organs to support cancer cell colonization [98]. Horizontal transfer of exosomal cargo to recipient cells creates a hospitable environment by suppressing anti-tumor immune response, creation of cancer associated fibroblasts, increasing vascular leakiness, remodeling of the extracellular matrix, and fibronectin deposition that is required for the successful dissemination and colonization of invasive melanomas [1,98,104]. We demonstrated mGluR1 expressing melanoma cells release higher amounts of exosomes/microvesicles (Goydos and colleagues unpublished work). Interestingly, when mGluR1 expression or function was modulated by genetic and pharmacologically means it reduced the aggressiveness of mGluR1 melanoma exosomes, but it did not affect the number of exosomes released in mGluR1 positive cells [103]. Furthermore, mGluR1 positive melanoma exosomes enhanced the migratory, invasive, and anchorage-independent growth of recipient cells [103]. These results suggest that mGluR1 signaling affects cargo sorting into exosomes. 

### 3.3. Therapeutic Targeting of Metabotropic Glutamate Receptor 1 Expressing Melanomas

Taken together, results from studies by our group and others unveil the importance of mGluR1 signaling in melanoma development and progression. mGluR1 positive melanomas have elevated levels of glutamate in the tumor microenvironment contributing to the hyperactivation of the receptor and its downstream effectors (Figure 4) [66]. We hypothesize that reducing extracellular glutamate in the microenvironment could be a viable therapeutic target. We took advantage of the known mGluR1 functions in the neuronal system and evaluated several well-known inhibitors of mGluR1 functions. We identified riluzole, an FDA-approved therapy for Amyotrophic Lateral Sclerosis (ALS) patients as the best candidate for anti-tumor activity. Riluzole is known to reduce glutamate export, thus acting as a functional inhibitor for mGluR1 in limiting the availability of extracellular glutamate to stimulate the receptor [66]. Inclusion of riluzole in the growth media of mGluR1 expressing melanoma cells led to decreased cell growth in vitro and reduced tumor progression in vivo with increased apoptotic cell population, with no obvious toxicities [66,105]. Human epidermal melanocytes are not affected by riluzole treatment [66]. We translated these laboratory findings to the clinic with a Phase 0 clinical trial followed by a Phase II single agent riluzole clinical trials in late-stage melanoma patients using the FDA approved dose of riluzole for ALS patients [70,71]. mGluR1 expression was not a criterion to participate in the trials, surprisingly, all patients on both trials were mGluR1 positive [70,71]. Significant decreases in FDG-PET scans, MAPK and PI3K/AKT signaling cascades and stable disease were achieved in 46% of patients, with little toxicity (dizziness, and dry mouth) suggesting that single agent riluzole is unlikely to have a long-lasting benefit in melanoma patients [71]. Treatment of cultured mGluR1-positive human melanoma cells showed the cells accumulated at the G2-M phase by 24 h and by 48 h there was an increased subG1 cell population suggesting apoptotic cells [66]. Cells arrested at the G2/M phase is indicative of cellular response to DNA damage, in line with this we detected elevated levels of γ-H2AX, a marker of DNA double-stranded breaks only in riluzole treated mGluR1 positive melanoma cells [106,107]. We speculate that riluzole interrupts glutamate export via the glutamate/cystine antiporter, xCT [106,108]. This leads to a decrease in the exchange of glutamate and cystine, resulting in a reduction in cysteine (the reduced form of cystine) to participate in glutathione (GSH) synthesis and a rise in the reactive oxygen species (ROS) levels in the cells as evident by elevated γ-H2AX [106,108]. Additional analyses using human tumor specimens from the completed Phase 0 trial confirmed these observations [106].

Damaged DNA if not repaired correctly will contribute to the mutation burden of the cells. This unique property of riluzole makes it an ideal component in rendering a “cold tumor into a hot tumor” possibly due to an increase in neoantigen load allowing for the identification of the tumor and recruitment of cytotoxic T-cells. In line with this, our lab has shown that in melanoma cells, DNA damage is preferentially repaired by the error prone non-homologous end joining (NHEJ) DNA repair pathway, and inclusion of riluzole in the media led to a decrease in the NHEJ repair efficiency that may further contribute to the tumor mutational burden of melanoma cells [109]. In line with this, tumor biopsies obtained from our Phase II riluzole monotherapy clinical trial showed an increase in immune cell infiltrates in stable disease patients compared to progressive disease patients suggesting that combining riluzole with immune checkpoint blockade therapy might enhance the efficacy of either agent alone [71]. However, the clinical use of riluzole is characterized by extensive hepatic metabolism and an exceptionally high level of patient-to-patient variability in drug exposure, due to variable first pass elimination effects mediated by heterogeneous expression of the cytochrome P450 isoform CYP1A2 [110]. This is one of the reasons that many patients with ALS do not benefit from riluzole due to inter-patient variability. Such high variability makes it exceedingly difficult to establish the pharmacokinetic (PK)/pharmacodynamic (PD) relationships needed to develop well-optimized dosing regimens for riluzole treatment for melanoma. To circumvent the first pass metabolism by CYP1A2, a prodrug of riluzole, troriluzole, was developed. This approach will bypass first pass liver metabolism and allow for the unform exposure of riluzole amongst patients independent of CYP1A2 expression.

Our lab is currently evaluating the therapeutic effects of troriluzole plus anti-PD-1 in our *Grm1* driven spontaneous melanoma prone mouse model, this combinatorial approach is also under clinical investigation in melanoma patients with brain metastases (NCT04899921). In our model, *Grm1* expression drives melanocyte hyperplasia that leads to nevi development, and progression into primary melanoma then metastatic melanoma (Figure 2 and Figure 3). The latency of onset and progression of melanoma is dependent on the number of copies of the disrupted *Grm1* gene (Figure 3B). Homozygous mice which harbor two copies of the disrupted *Grm1* gene display raised pigmented lesions by four to six weeks and succumb to high tumor burden by four months of age (Figure 3B). Heterozygous mice which harbor only one copy of the disrupted *Grm1* gene shows very similar onset and progression of melanoma to the homozygous mice but raised lesions occur at a much later time and they succumb to high tumor burden by ten to twelve months of age (Figure 3B). These mice develop spontaneous melanoma without any chemical or UV induction; therefore we propose that the tumor-stromal interactions as well as the tumor-immune system interactions are physiologically relevant to human melanomas. This notion was supported by the report that these mice have similar immune system dysfunctions as human melanoma patients and is one of the contributors to melanoma development [111,112,113,114]. Further discussion of the immune dysfunction occurring in our *Grm1*- driven melanoma mice have been reviewed by Eddy et al., in “Overcoming Immune Evasion in Melanoma” [1]. Taken together these data suggest that our spontaneous melanoma prone mouse model recapitulates various stages of human melanoma and represent a clinically relevant model to accurately predict treatment response in human melanomas.

### 3.4. Regulation of Metabotropic Glutamate Receptor 1 in Melanocytes and Melanoma Cells

mGluR1 expression has been detected in neuronal and non-neuronal cells [115,116]. We uncovered the oncogenic properties of mGluR1 when mGluR1 was ectopically expressed in melanocytes, suggesting that mGluR1 expression is highly regulated. We explored and compared the transcription machineries found in normal human melanocytes and human melanoma cells. Earlier studies by others identified the Neuron Restrictive Silencing Factor (NRSF) as a master regulator of neuronal specific gene expression in non-neuronal cells. NRSF, a Kruppel-type zinc finger transcription factor that interacts with 23 base pair cis-acting Neuron Restrictive Silencer Element (NRSE) suppress neuron-specific gene expression in non-neuronal cells [117,118]. Ferraguti and co-workers identified a consensus NRSE site within 5 kb upstream of the *Grm1* initiating codon, in addition, the lack of mGluR1 expression in BHK and NIH3T3 cells was mediated by the binding of NRSF to NRSE [115]. Similar results were observed in normal human epidermal melanocytes, however, in melanoma cells in addition to the interactions of NRSF and NRSE, another well-known transcription factor, Sp1 plus methylation at the *GRM1* promotor region also appear to participate in the regulation of *GRM1* expression in melanocytes [116]. These results unveil the complex and tightly regulated nature of *GRM1* expression in human melanocytes and melanomas. To identify additional regulators, a high throughput unbiased proteomic and regulatory RNA screen within the *GRM1* promoter and enhancer regions between human melanocytes, paired mGluR1 positive tumors (and/or cell lines) and paired mGluR1 negative tumors (and/or cell lines) are needed. Using reverse-chromatin immunoprecipitation (r-CHIP) a pull down of the *GRM1* promoter and enhancer regions followed by mass spectrometry will identify proteins bound in these regions across the samples described above [119,120]. As stated previously a 70 kb genomic region was deleted with the insertion of the transgene within intron 3 of the *Grm1* gene in the TG-3 mice (Figure 2A) [64]. We hypothesis that this deleted 70 kb region may contain regulatory elements such as non-coding RNA or the deletion plus the insertion in intron 3 of *Grm1* modified the accessibility of chromatin remodeling complexes and/or transcription factors that may promote *Grm1/GRM1* transcription.

### 3.5. Implications of Other Metabotropic Glutamate Receptors in Melanomas

In addition to mGluR1 other mGluRs have also been shown to drive glutamatergic signaling in melanomas such as mGluR3, mGluR5, mGluR8 and ionotropic glutamate receptor (iGluR/GRIN2A) (Table 1) [40,121,122,123,124,125,126,127]. Overexpression of mGluR5 has been implicated in melanoma development and progression in both mouse and humans, however mGluR5 is normally expressed in both normal melanocytes and melanoma tumors (Table 1) [121,128]. Like TG-3, mGluR5 overexpression in melanoma cells leads to hyperactivation of the MAPK pathway, as shown by enhanced levels of phosphorylated ERK in TRP1-*Grm5* driven melanoma mouse model compared to the wild-type littermates [121]. We demonstrated earlier that mGluR5 is not required for the oncogenic activities of *Grm1* mediated transformation of melanocytes into malignant melanoma [81]. Alterations in the regulation of mGluR1 and mGluR5 expression are the key mediators in transformed melanocytes and tumor formation, while other mutations in mGluR3, mGluR8 and iGluR are implicated in melanomagenesis (Table 1) [40,122,123]. Activating mutations within *GRM3* leads to the hyperactivation of the MAPK pathway, which corresponds with an increase in cell migration, proliferation, and anchorage-independent growth (Table 1) [123]. A recent report by Ceol and colleagues demonstrated that melanoma associated mGluR3 variant signaling downregulates cAMP signaling that affects melanosome trafficking that may contribute to melanomagenesis from cross talks between cAMP and MAPK signaling cascades and lead to drug resistance, however, further validation is needed [124]. Somatic mutations in iGluRs were shown to increase the proliferative and metastatic capacities of melanoma cells [125,126]. A study by D’Mello et al., demonstrated mutations in iGluRs are associated with worse survival, and faster disease progression for late-stage melanoma patients, but the sample size was small and has to be further validated in a larger cohort [127]. The specific function of mGluR8 in melanoma pathogenesis has yet to be characterized, and we suspect it may have similar function as mGluR3 as both mGluR3/mGluR8 modulate cAMP signaling (Table 1) [129].

## 4. Involvement of Glutamatergic Signaling in Other Cancers

Glutamatergic signaling has been implicated in neuronal tumors, breast cancers, kidney cancers, colorectal cancers, gastrointestinal cancers, and prostate cancers (Table 1) [40,136,137,138,139,140,141,142,143,144,145,146,147,148,149,151,154,155,156,158,159,160]. mGluR1, mGluR2, and mGluR3 signaling have been implicated in promoting glioma cells tumorigenicity and metastatic potentials via activated MAPK and PI3K/AKT pathways (Table 1) [144,151,152,153,160]. In a set of studies with glioma cells, we demonstrated elevated levels of glutamate in the tumor microenvironment of mGluR3 positive glioma cells. Like our results in mGluR1 positive melanoma cells, riluzole treatment of mGluR3 positive glioma cells led to a reduction in glioma cell growth in vitro and tumor progression in vivo [153]. Combining riluzole with γ-irradiation enhanced the cytotoxic effects of both agents as reported earlier [106,107,153]. Furthermore, riluzole and its prodrug, troriluzole can cross the brain-blood barrier and promote oxidative stress resulting in DNA damage in targeted cells, rendering them more sensitive to γ-irradiation. This approach is a relatively non-toxic approach to enhance the therapeutic activities of γ-irradiation in the treatment of gliomas, brain metastases, and other neurological diseases.

The oncogenic effects of *Grm1* in mammary tumors were shown using immortalized mouse mammary epithelial cells (iMMECs) [133]. Introduction of *Grm1* cDNA into iMMECs induced cellular transformation in vitro and tumor formation in vivo with enhanced tumor angiogenesis [133]. Sustained mGluR1 signaling in breast cancers is necessary in tumor angiogenesis, tumor progression, and tumor promoting inflammation (Table 1) [132,133,161,162]. In estrogen receptor (ER) positive, ER negative and triple negative breast cancers (TNBC), mGluR1 expression is a valuable prognostic marker in predicting patient survival [131,134,135]. These findings prompted our group and the Gorski group to evaluate the efficacy of riluzole in mGluR1 expressing breast cancers [130,163,164]. Riluzole treated mGluR1 expressing breast cancers had reduced cell growth and viability in vitro and tumor progression in vivo [130,163,164]. Surprisingly, the efficacy of riluzole is independent of mGluR1 expression in breast cancer cells and is proposed to be responsible for alterations in cellular metabolism by inhibition of de novo pyrimidine synthesis and oxidative phosphorylation [163,164]. These results may be attributed to the fact that riluzole mediates its activities via xCT and mGluR1 negative breast cancer cells may be sensitive to shifts in reduction-oxidation (REDOX) states [108,163,164]. Additionally, mGluR4 has also been implicated in breast cancer, where it acts as a tumor suppressor gene rather than an oncogene (Table 1) [157]. Ectopic expression of mGluR4 was responsible for improved patient prognosis and biological assays revealed a reduction in the proliferative and metastatic capacities of breast cancer cells expressing mGluR4 [157].

Introduction of Grm1 cDNA into immortalized primary baby mouse kidney (iBMK) cells resulted in cellular transformation in vitro and tumor formation in vivo [145]. Sustained mGluR1 signaling through the MAPK and PI3K/AKT pathways is required to maintain tumorigenic phenotypes in mGluR1 expressing iBMK cells [145]. These findings prompted us to examine the importance of glutamatergic signaling in human renal cell carcinoma (RCC). All human RCC cell lines tested were positive for mGluR1 expression (Table 1) [145]. In agreement with our previous work on melanoma and breast cancer cells, RCC cells that express mGluR1 also release excess glutamate into extracellular environment and are sensitive to riluzole treatment supporting the notion that sustained mGluR1 signaling is required for tumor progression [145]. A recent report has suggested that the genetic variants of *GRM3*, and *GRM4* in RCC are associated with worser survival while *GRM5* is a risk factor for developing RCC (Table 1) [150].

Colorectal carcinoma is a cancer that affects the colon and rectum and was shown to be driven by glutamatergic signaling via mGluR4 overexpression (Table 1) [155]. Clinical data reveals that overexpression of mGluR4 is correlated with tumor recurrence and poor disease survival [155]. Another cancer that mGluR1 has been implicated in is prostate cancer (Table 1) [146]. Primary and metastatic prostate cancer specimens and cell lines were shown to express mGluR1 while normal prostate cells showed little or undetectable mGluR1 expression [146]. Riluzole treatment abrogated prostate cancer cell growth, induced apoptosis, and decreased metastatic capabilities of these cells [146].

## 5. Conclusions and Future Perspectives

In recent years, important advances in our mechanistic understanding of melanoma biology have advanced patient care that has resulted in the development of targeted therapies and immunotherapies. These therapies although successful are marked by resistance to treatment, disease relapse and only benefit some patients, suggesting that we still have much to learn and understand about the complex signaling networks in melanomas and other cancers.

Data from our lab and others have demonstrated that glutamatergic signaling mediated by mGluRs may be a therapeutic viable target for treating cancer patients. mGluR1 when abnormally expressed or harboring activating mutations can promote melanocytes to transform into malignant melanoma by hyperactivating the oncogenic pathways MAPK and PI3K/AKT pathway suggesting mGluR1 may be upstream of many signaling pathways (Figure 4). mGluR1 in melanoma appears to participate in many aspects of transformation including cell growth, metabolism, metastasis, angiogenesis, and survival via the establishment of an autocrine/paracrine loop that results in abundant amounts of glutamate in the tumor microenvironment to ensure constitutive activation of the receptor (Figure 4 and Figure 5). Inclusion of a relatively non-toxic reagent, riluzole or its prodrug troriluzole has profound anti-tumor activity. We hypothesize that xCT, a glutamate-cystine antiporter plays a vital role in the mode of action of riluzole/troriluzole. All cells utilize the xCT antiporter to export one molecule of glutamate to the outside the cell and import one molecule of cystine into the cell (Figure 4). Once in the cell, cystine is reduced to cysteine, which is then used for the synthesis of GSH. GSH neutralize ROS. If the concentrations of ROS exceed the limits of melanoma cells (or other cancers) it will hinder their fitness, therefore it is important for cancer cells to maintain an appropriate REDOX state. To maintain REDOX homeostasis in cancer cells, xCT is frequently upregulated to allow for the appropriate exchange of cystine with glutamate. In mGluR1 expressing melanoma cells upregulated xCT results in enhanced receptor activities as a result of ample levels of its natural ligand, glutamate, rendering the cells addicted to glutamate and reprogramming of cellular metabolisms (Figure 4). Increased glutaminase (GLS) which catalyzes the deamination of glutamine to glutamate before entering the TCA cycle, was noted in mGluR1 expressing melanoma cells (Figure 4) [80,108,165]. Furthermore, we also showed a correlation between mGluR1 and GLS expression (Figure 4). Treatment of mGluR1-positive melanoma cells with an inhibitor of GLS, CB-839, leads to a reduced glutamate pool and decreased viable melanoma cells in vitro and in vivo [80]. The interactions between GLS, xCT and mGluR1 may support tumor promoting inflammatory response and may be responsible for establishing an immune suppressive shield around the tumor. Furthermore, riluzole/troriluzole treatment may induce DNA damage, promoting mutagenesis, thereby making mGluR1 tumor cells more immunogenic and responsive to immunotherapy agents.

Various reports have shown that the NLR family pyrin domain containing (NLRP) subfamily member 3 (NLRP3) is the major sensor for intracellular danger signals [166]. Activation of NLRP3 by cellular stress results in inflammasome activation catalyzing the formation of proinflammatory cytokines, interleukin IL- 1β and IL-18 [166,167]. Inflammasomes are upregulated in melanomas and NLRP3 expression and activity increases as melanomas progress [166,167,168]. In the tumor microenvironment, an increased in IL-1β and IL-18 led to decreased B-cells, CD8+ T-Cells, CD4+ T-cells, macrophages, neutrophils, and dendritic cells [166,168,169]. Further evidence suggests that NLRP3/IL-1 activity in melanoma cells, contributes to the expansion of myeloid derived suppressor cells (MDSCS) in the tumor microenvironment that corresponds with a reduction in NK- and CD8^+^ T-cell activity and increase in regulatory T-cells (T-Regs) [169]. Targeting of NLRP3 by, OLT1177, was shown to reduce MDSC recruitment to the primary tumor, enhanced anti-tumor immune response and bolstered anti-PD-1 treatment in vivo [169]. Quagliariello and colleagues demonstrated in breast cancer cells expressing NLRP3, that NLRP3 is associated with insensitivity to Ipilimumab (anti-CTLA-4), but by targeting NLRP3 with OLT1177, the cytotoxicity of Ipilimumab is increased, and a reduction of cardiotoxicity is observed, a common immune related adverse effect associated with immune checkpoint therapy [170]. NLRP3 regulates the inflammatory response in cells, and chronic inflammation has been implicated with the risk of melanoma, it is possible that factors, which increase the risk of inflammation such as obesity, hyperglycemia and/or diabetes, may result in melanoma development and progression via NLRP3 [170,171,172,173]. These results suggest that melanoma patients who are obese, hyperglycemic, and/or diabetic may consider adding NLRP3 inhibition or metformin, a commonly used treatment for, controlling diabetes or hyperglycemia, while undergoing treatment for melanoma [174]. Recent reports on the preclinical development of highly precise radioactive probes/pharmaceuticals that target mGluR1 suggest that these could be a valuable tool for the diagnoses and treatment of mGluR1 positive disorders [175,176].

## Figures and Tables

**Figure 1 cancers-13-03874-f001:**
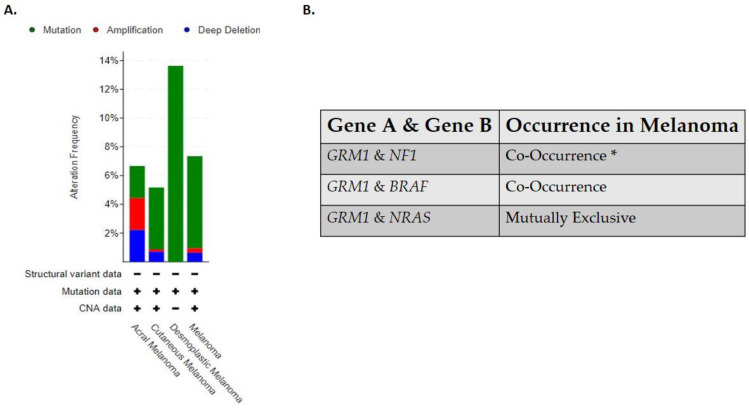
(**A**) Alterations in *GRM1* that are found in human melanomas. (**B**) Occurrence of *GRM1* with other melanoma driver genes in human melanomas. Analyses were done using cBioPortal.org (accessed on 15 July 2021). The skin dataset was chosen under query. All melanoma studies were included in the analyses, except Skin Cutaneous Melanoma (TCGA, PanCancer Atlas) due to the dataset overlapping with Skin Cutaneous Melanoma (TCGA, Firehouse Legacy). The sample size was 2386. * Significant.

**Figure 2 cancers-13-03874-f002:**
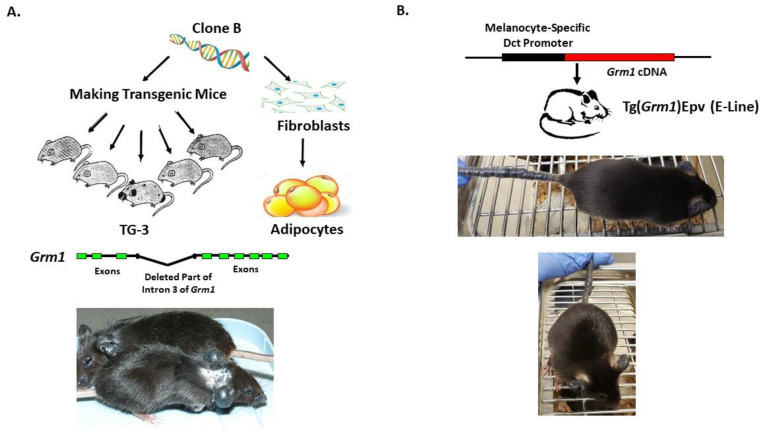
(**A**) Derivation of TG-3 and (**B**) Tg(*Grm1*)Epv Spontaneous *Grm1* Expressing Melanoma Prone Mouse Models.

**Figure 3 cancers-13-03874-f003:**
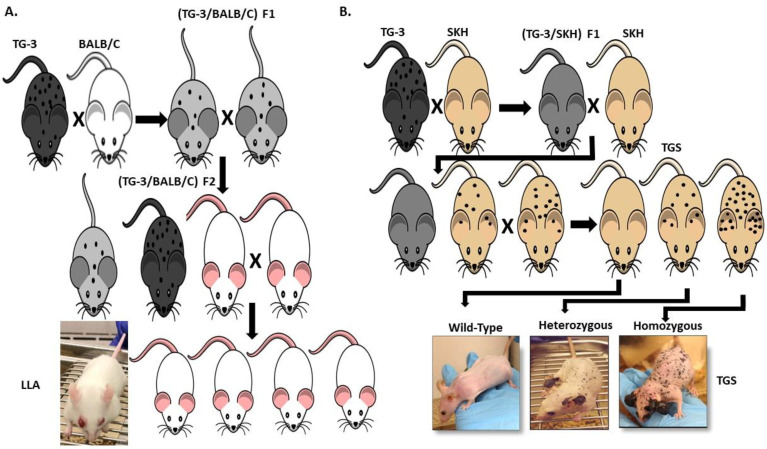
(**A**) Derivation of a Spontaneous *Grm1* Expressing Amelanotic Melanoma Prone Mouse Model, LLA and (**B**) Hairless Spontaneous *Grm1* Expressing Melanoma Prone Mouse Model, TGS.

**Figure 4 cancers-13-03874-f004:**
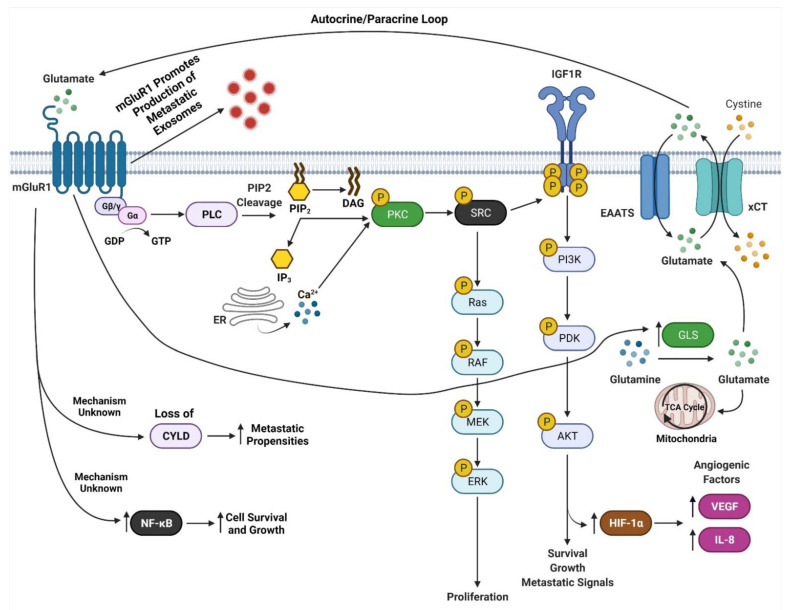
mGluR1 Signaling in Melanoma. Adapted from “Glutamate Signaling Pathways in Melanoma and Potential Therapeutic Intervention Points”, by BioRender.com (accessed on 1 July 2021). Retrieved from https://app.biorender.com/biorender-templates (accessed on 1 July 2021).

**Figure 5 cancers-13-03874-f005:**
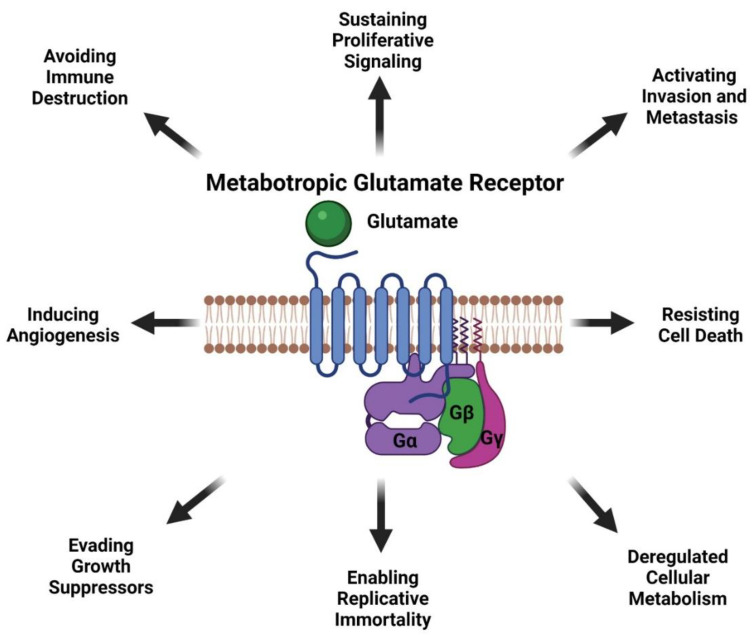
Oncogenic Activities of Metabotropic Glutamate Receptor in Melanoma. Created with BioRender.com (accessed on 1 July 2021).

**Table 1 cancers-13-03874-t001:** mGluRs in Various Malignancies.

mGluRs	Malignancies	References	Ref Number
**mGluR1**	Malignant Melanoma	Pollock et al., 2003; Ohtani et al., 2008	[64,67]
	Breast	Speyer et al., 2012; Mehta et al., 2013; Banda et., 2014; Teh et al., 2015; Stires et al., 2018; Bastiaansen et al., 2020	[130,131,132,133,134,135]
	Lung	Kan et al., 2010; Esseltine et al., 2013	[136,137]
	Ovary	Cancer Genome Atlas Research, 2021	[15]
	Large Intestine	Sjoblom et al., 2006; Cancer Genome Atlas Research, 2021	[15,138]
	Upper Aerodigestive Tract	Durinck et al., 2011; Stransky et al., 2011; Esseltine et al., 2013	[137,139,140]
	Astrocytoma	Parsons et al., 2008	[141]
	Glioma	Stepulak et al., 2009; Brocke et al., 2010; Zhang et al., 2015	[142,143,144]
	Medulloblastoma	Brocke et al., 2010	[143]
	Renal Cell Carcinoma	Martino et al., 2013	[145]
	Prostate	Koochekpour el al., 2012	[146]
**mGluR5**	Malignant Melanoma	Frati et al., 2000; Choi et al., 2011	[121,128]
	Prostate	Pissimissis et al., 2009	[147]
	Oral Squamous Cell Carcinoma	Park et al., 2007	[148]
	Osteosarcoma	Kalariti et al., 2007	[149]
	Glioma	Stepulak et al., 2009; Brocke et al., 2010	[142,143]
	Medulloblastoma	Brocke et al., 2010	[143]
	Renal Cell Carcinoma	Huang et al., 2018	[150]
**mGluR2**	Glioma	D’Onofrio et al., 2003; Arcella et al., 2005; Stepulak et al., 2009	[142,151,152]
	Prostate	Pissimissis et al., 2009	[147]
**mGluR3**	Glioma	D’Onofrio et al., 2003; Arcella et al., 2005; Ciceroni et al., 2008; Stepulak et al., 2009; Prickett et al., 2011; Khan et al., 2019	[123,142,151,152,153]
	Malignant Melanoma	Prickett et al., 2011; Prickett and Samuels., 2012; Neto et al., 2018	[123,124,154]
	Renal Cell Carcinoma	Huang et al., 2018	[150]
**mGluR4**	Colorectal Carcinoma	Chang et al., 2005	[155]
	Glioma	Arcella et al., 2005; Stepulak et al., 2009; Brocke et al., 2010	[143,152]
	Malignant Melanoma	Chang et al., 2005	[155]
	Squamous Cell Carcinoma	Chang et al., 2005	[155]
	Medulloblastoma	Iacovelli et al., 2006	[156]
	Breast	Xiao et al., 2019	[157]
	Renal Cell Carcinoma	Huang et al., 2018	[150]
**mGluR6**	Glioma	Arcella et al., 2005; Stepulak et al., 2009; Brocke et al., 2010	[143,152]
	Medulloblastoma	Brocke et al., 2010	[143]
**mGluR7**	Glioma	Stepulak et al., 2009	[142]
**mGluR8**	Malignant Melanoma	Choi et al., 2011; Prickett and Samuels, 2012	[121,154]
	Glioma	Stepulak et al., 2009	[142]

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
