# Peer review of "Glutamatergic Signaling a Therapeutic Vulnerability in Melanoma"

_cancers, 2021, doi:10.3390/cancers13153874_

Round 1

Reviewer 1 Report

In this manuscript, Eddy and Chen review the role of Metabotropic Glutamate Receptor 1 (GRM1) in melanoma development. They provide an overview of melanoma genetics before introducing the mouse models of Grm1-driven melanoma reported in their seminal 2003 paper. They describe the various consequences of mGluR1 signaling that could contribute to melanoma initiation and progression, including MAPK and PI3K pathway activation, angiogenesis, exosome release, and DNA damage. They present various studies on the use of riluzole, an inhibitor of glutamate export, as an anti-cancer agent. Finally, they broaden the scope of their review by mentioning works implicating other glutamate receptors in malignancies other than melanoma.

Although mostly centered on work done by their own research group, this manuscript gives a rather comprehensive view of the current knowledge on glutamatergic signaling in melanoma. Addressing the following points would strengthen the content or improve the overall readability of the paper.

Major points:

1) Authors insist on melanoma genetics. However, it is unclear if there are genetic alterations in GRM1 in human melanoma and, if so, how frequent these alterations are. It would be useful to add a diagram showing the cooccurrence or mutual exclusivity of GRM1 alterations with alterations in the established melanoma driver oncogenes BRAF, NRAS and NF1.  

2) The authors rightly present metastasis as the main cause of melanoma-related death. They mention the potential role of mGluR1 in promoting melanoma aggressiveness but do not develop experimental arguments to support this claim. Are mGluR1-positive melanomas more metastatic? Is there a causal link between mGluR1 expression or activity and tumor dissemination?

3) The description of studies using riluzole to treat cancer is informative. The authors should mention any toxicities associated with these treatments. Have the effects of this compound on non-melanoma cells (and normal cells, including melanocytes) been evaluated?

4) The authors provide strong evidence that mGluR1 overexpression can initiate melanoma formation and promote tumor progression. Is there any data supporting a role for mGluR1 in melanoma maintenance, i.e. suggesting that genetic or pharmacologic inhibition of GRM1/mGluR1 would block melanoma cell proliferation or induce melanoma cell death in a non-Grm1-driven model?

5) The wording of a whole paragraph (from line 103 to line 122) is somewhat awkward and needs rewriting.

Minor points:

1) Breaking down part 3 into several subparts corresponding to the various aspects of GRM1 biology (clinical data, DNA repair, mutation burden, transcriptional regulation, etc) and adding subheadings would improve the readability of the paper.

2) In their introduction (parts 1 or 2), the authors should mention the fact that a significant proportion (maybe even a majority) of melanomas do not develop from nevi.

3) Lines 76-77 “Like uveal melanoma, mucosal melanoma also harbors mutations in the GNAQ/GNA11 genes”. This sentence is misleading. GNA11/GNAQ mutations are rare in mucosal melanoma whereas they represent the main drivers of uveal melanoma. According to the most recent genomic studies of mucosal melanoma, the main drivers of MAPK activation in this melanoma subtype appear to be KIT mutations, NF1 loss-of-function or SPRED1 loss-of-function alterations. The authors should clarify this point.

4) Line 194 “melanocytes are of neuronal origin”. Change to “neural origin”. Melanocytes derive from the neural crest.

5) Please explain what CYLD is. Is there evidence that CYLD is downregulated in mGluR1-positive human melanoma cells (compared to GRM1-negative cells)?

6) Can the authors provide some hypotheses regarding the fact that some effects of riluzole seem independent of mGluR1 expression?

7) Please consider revising the last sentence of the abstract since the mechanisms regulating mGluR1 expression are only briefly discussed whereas most of the review focuses on the consequences of mGluR1 overexpression.

Reviewer 2 Report

The manuscript titled "Glutamatergic Signaling a Therapeutic Vulnerability in Melanoma" is an interesting review describing the role of some pathways able to modulate the therapeutic efficacy in melanoma. The inreoduction is well organized and the overall structure of the manuscript is of good quality. Referenes are updated and of good quality; Authors should explain the role of other dietetic determinants on the anticancer efficacy of ICIs in melanoma or other cancers like hyperglycemia ( you can cite doi: 10.3390/ijms21207802 ) and the role of NMDA related pathways in IL-1 mediated resistance. Moreover, the authors should describe the role of NLRP3 in gluraminergic pathways related to pharmacological responsiveness in melanoma patients. 

The manuscript is acceptable after minor revision

Reviewer 3 Report

The paper is well written. As it is more focused on the preclinical pathways it is difficult for me to review. This is not my expertise.

Some small remarks: Sentences "Our lab was the first to show the oncogenic properties of ectopic mGluR1 expression in melanoma development and 162 progression. ... that was shown to commit fibroblasts to un-164 dergo adipocyte differentiation in vitro (Figure 1A)" refer to papers published decades ago. What is the relevance for the current knowledge/review? 

A lot of references has been used, but it is not a systematic review. There is no method section thus I cannot assess whether all relevant literature is included. 
